# Beam-induced motion correction for sub-megadalton cryo-EM particles

Sjors HW Scheres*

Structural Studies, Medical Research Council Laboratory of Molecular Biology, Cambridge, United Kingdom

**Abstract** In electron cryo-microscopy (cryo-EM), the electron beam that is used for imaging also causes the sample to move. This motion blurs the images and limits the resolution attainable by single-particle analysis. In a previous Research article (*Bai et al., 2013*) we showed that correcting for this motion by processing movies from fast direct-electron detectors allowed structure determination to near-atomic resolution from 35,000 ribosome particles. In this Research advance article, we show that an improved movie processing algorithm is applicable to a much wider range of specimens. The new algorithm estimates straight movement tracks by considering multiple particles that are close to each other in the field of view, and models the fall-off of high-resolution information content by radiation damage in a dose-dependent manner. Application of the new algorithm to four data sets illustrates its potential for significantly improving cryo-EM structures, even for particles that are smaller than 200 kDa.

## Introduction

The recent development of highly efficient direct-electron detectors and powerful new image processing algorithms has led to rapid progress in the resolutions obtained by electron cryo-microscopy (cryo-EM) structure determination. The new detectors yield higher signal-to-noise ratios (SNRs) than conventional detection devices (*McMullan et al., 2009*), and are fast enough to record multiple images, that is movies, during typical exposure times. The latter allows correction for sample movements that are caused by interactions with the incoming electron beam (*Brilot et al., 2012*; *Campbell et al., 2012*). Last year, we reported an algorithm for beam-induced movement correction of individual ribosome particles, which allowed us to calculate a reconstruction with a level of detail down to 4 Å resolution from only 35,000 (asymmetric) ribosome particles (*Bai et al., 2013*). Around the same time, *Li et al. (2013)* reported an algorithm to correct for the movement of much larger fields of views, which they used to calculate a 3.3 Å map for the 20S proteasome from 1.8 million asymmetric units. More recently, these techniques were also used to calculate a 3.2 Å map for the large subunit of the yeast mitochondrial ribosomal subunit (*Amunts et al., 2014*), a 3.4 Å structure of the $F_{420}$-reducing [NiFe] hydrogenase (*Allegretti et al., 2014*), and a 3.4 Å structure of the TRPV1 ion channel (*Liao et al., 2013*).

This paper describes an advance on our previously described beam-induced movement correction algorithm. The original algorithm followed particle movement by determining the relative rotations and translations of individual particles in running averages of several movie frames with respect to a common reference map. However, this approach does not work well for particles that are much smaller than ribosomes, because too low SNRs in the movie frames prohibit accurate orientation determination. Although the approach by *Li et al. (2013)* suffers less from lower SNRs in smaller particles (because each field of view contains many of them), it is less suited to model the complicated movement patterns that we and others have observed (*Glaeser and Hall, 2011*; *Brilot et al., 2012*; *Bai et al., 2013*) (also see *Figure 1*). In addition, neither of the two algorithms accounts for the observation that low frequencies in the data survive higher electron doses than the high frequencies (*Unwin and Henderson, 1975*;

*For correspondence: scheres@mrc-lmb.cam.ac.uk

**Reviewing editor**: Werner Kühlbrandt, Max Planck Institute of Biophysics, Germany

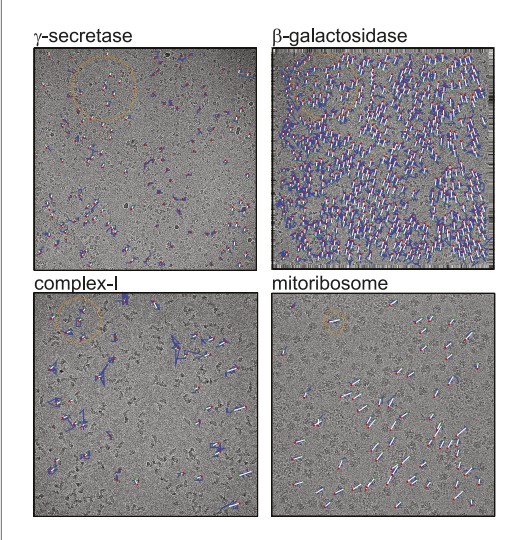

**Figure 1**. Beam-induced movement tracks. A representative micrograph for each of the four test cases is shown, on top of which 50-fold exaggerated beam-induced particle movements are plotted. The original tracks as estimated for running averages of several movie frames for each particle independently are shown in blue; the fitted linear tracks are shown in white. The start and end points of the fitted tracks are indicated with green and red dots, respectively. The orange circles indicate the $2\sigma_{NB}$ distance for one of the particles on the micrographs. Note that tracks are only shown for those particles that were selected for the final reconstruction after 2D and 3D classification. Also note that the relatively small movement tracks for γ-secretase only represent the beam-induced motion that was not already corrected for in the algorithm by *Li et al. (2013)*.

*Hayward and Glaeser, 1979*; *Stark et al., 1996*). The approach presented here addresses both issues: it deals with relatively small particles by considering the movements of multiple particles together, yet is still able to model complex movement patterns, and it formulates a resolution-dependent radiation damage model.

## Approach

In our previous contribution (*Bai et al., 2013*), we observed beam-induced rotations for ribosome particles of only a few degrees, which is comparable to the accuracy with which orientations of the individual particles could be determined. This agreed with previous observations by others (*Campbell et al., 2012*), and prompted us to investigate a beam-induced movement correction algorithm that only considers particle translations. The new approach builds on the original one, where running averages of several movie frames of individual particles are (only translationally) aligned against a common reference. Because for small particles these translations become noisy, the new approach fits straight tracks through them, and includes all neighbouring particles on the same field of view in these weighted least-squares fits. More specifically, for each particle $p$, the program independently minimizes the least squares target below for both the X- and Y-components of the particle coordinates in the field of view:

$$\min_{\alpha_p, \beta_p} \left( \sum_{p'} w_{p'} \sum_f \left( \Delta x_{p'} - \alpha_p - \beta_p f \right)^2 \right),$$

where the first summation runs over all particles $p'$ in the field of view; $w_{p'}$ is calculated as a Gaussian function of the distance between particles $p$ and $p'$; the second summation runs over all movie frames $f = 1,…,F$; $\Delta x_{p'}$ is the difference in (either X- or Y-) coordinate between the refined position of the average particle $p'$ and the refined position of its movie frame $f$; and $\alpha_p$ and $\beta_p$ are the intercept and slope parameters of the fitted movement track for particle $p$ in the X- or Y-direction. The user controls the standard deviation $\sigma_{NB}$ of the Gaussian function that defines all $w_{p'}$. Smaller values for $\sigma_{NB}$ result in fewer neighbouring particles effectively contributing to the linear fits, so that less noise may be removed, but more complicated movement patterns may be described. In cases where the beam-induced motions may not be modelled by straight tracks, for example for bent or zig-zag tracks, this method will not yield good results. In such cases, a hybrid approach, as also used for the γ-secretase example below, may be useful to correct for large, possibly non-straight movements using the algorithm by *Li et al. (2013)* and correct for finer movements using the one described here.

The fall-off of information content with frequency in cryo-EM data is often modelled by a Gaussian using a B-factor, in analogy to the temperature factor in X-ray crystallography (*Rosenthal and Henderson, 2003*). The new movie processing approach exploits a similar model to describe the dose-dependent signal fall-off throughout the recorded movies by estimating a (relative) B-factor for each movie frame. The most common procedure to estimate B-factors in cryo-EM structure determination, by analyzing the Guinier plot of the logarithm of the amplitudes ($\bar{l}$) of a reconstruction versus the square of the frequency ($v^2 = 1/\delta^2$), only works for reconstructions where the signal extends beyond

$v$ = 1/10 Å⁻¹ (**Rosenthal and Henderson, 2003**). Because reconstructions from individual movie frames often do not reach that resolution, an alternative analysis was devised. Instead of estimating absolute B-factors for each movie frame, the alternative approach merely estimates 'relative B-factors' that describe how much faster (or slower) the signal in reconstructions from individual frames drops with frequency compared to the average reconstruction from all movie frames. If the signal in a reconstruction from a single movie frame of all particles drops faster than the signal in a reconstruction from all movie frames, then the relative B-factor will be negative; if the signal in the single-frame reconstruction drops slower, the relative B-factor will be positive. Assuming equal noise power spectra ($\sigma^2$) in all single-frame reconstructions, the ratio of the amplitudes of the signal in reconstructions from each individual movie frame ($f$) versus the average reconstruction from all movie frames ($a$) may be calculated as:

$$\frac{\tau_f(v)}{\tau_a(v)} = \sqrt{\frac{FSC_f(v) - FSC_f(v)\,FSC_a(v)}{FSC_a(v) - FSC_f(v)\,FSC_a(v)}},$$

where $FSC_f$ and $FSC_a$ are Fourier shell correlation curves calculated between two independently refined halves of the data for the individual frame and average reconstructions; and one uses $\tau^2(v)/\sigma^2(v)$ = $SNR(v)$ = $FSC(v)/\{1 - FSC(v)\}$. Plotting the logarithm of $\tau_f(v)/\tau_a(v)$ versus the square of the frequencies then produces a 'relative Guinier' plot, that is, it describes the fall-off of signal of an individual movie frame relative to the average reconstruction from all movie frames. Fitting straight lines through the relative Guinier plot can then be used to derive the relative B-factors, $B_f$, and the intercepts of the fitted lines with the Y-axis, $C_f$, which represent frequency-independent components of the relative signals. In these plots, useful frequencies for fitting straight lines were observed from approximately 1/20 Å⁻¹ to the frequency where $FSC_f$ = 0.143. The intercept $C_f$ and the slope $B_f$ of these fits are then used to calculate a frequency-dependent weight, $w_f$, for all movie frames as:

$$w_f(v) = \frac{\exp\left(\frac{B_f}{4}v^2 + C_f\right)}{\sum_{f'} w_{f'}(v)}.$$

Finally, because out-of-plane rotations are not considered in the new approach, one may apply the in-plane translations from the fitted tracks for all movie frames of each particle, and then sum the movie frames using the frequency-dependent weights described above. This creates a set of new 'polished' average particles with increased SNRs, which may again be used in subsequent 2D or 3D classifications or refinements. It is useful to consider that, since all Guinier plots are calculated relative to the same average reconstruction, the absolute values of $B_f$ and $C_f$ do not matter in the calculation of the weighted average particles: only the differences between these values for different movie frames determine the relative contribution of each movie frame to each frequency shell ($v$). Also, because of the summation in the nominator above, the sum of all $w_f(v)$ over all movie frames remains unity within each frequency shell. Therefore, the re-weighting of the movie frames thus does not comprise an overall sharpening or dampening of the data, and the estimation of an overall absolute B-factor (as in the last row of **Table 1**) remains relevant.

## Results and discussion

The new approach was tested on four previously published cryo-EM data sets on particles of varying size: human γ-secretase (**Lu et al., 2014**), *Escherichia coli* β-galactosidase (**Scheres and Chen, 2012**; **Chen et al., 2013**; **Vinothkumar et al., 2014a**), bovine complex-I (**Vinothkumar et al., 2014b**), and the large sub-unit (LSU) of the yeast mitochondrial ribosome (**Amunts et al., 2014**). The data set on the γ-secretase complex was recorded on a Gatan K2-Summit detector; all other data sets were recorded on an FEI Falcon-II detector. Because the K2 detector aims to count single-electron events, it has to be used at a much lower dose rate than the Falcon-II, which integrates charge during the exposure. Consequently, the γ-secretase data were recorded using relatively long exposures of 15 s. To correct for linear drift during this time, the recorded fields of view were first subjected to the algorithm by **Li et al. (2013)**, and then to the particle-based correction described here. A similar hybrid strategy, but using the original movie processing approach, was also applied to F₄₂₀-reducing [NiFe] hydrogenase (**Allegretti et al., 2014**). The other three data sets were recorded with exposures of 1–2 s, and only particle-based movement correction was performed. The results are summarized in **Table 1**.

**Table 1.** Overview of the results

| | γ-Secretase | β-Galactosidase | Complex-I | Mitoribosome large sub-unit |
|---|---|---|---|---|
| Molecular mass (MDa) | 0.17* | 0.45 | 1.0 | 1.9 |
| Data set characteristics | | | | |
| Sample support | Quantifoil R1.2/1.3 | Quantifoil R1.2/1.3 | Quantifoil R0.6/1 | Quantifoil R2/2 continuous carbon |
| Microscope | Titan Krios | Polara | Titan Krios | Titan Krios |
| Detector | K2-Summit | Falcon-II | Falcon-II | Falcon-II |
| Pixel size (Å) | 1.76 | 1.77 | 1.71 | 1.34 |
| No. movie frames | 15 | 24 | 32 | 17 |
| Exposure time (s) | 15 | 1.5 | 1.9 | 1 |
| Electron dose (e$^-$/Å$^2$) | 37 | 24 | 32 | 25 |
| No. particles | 144,545 | 34,032 | 45,618 | 47,114 |
| Prior to movie processing | | | | |
| Resolution (Å) | 4.9† | 4.3 | 5.9 | 3.9 |
| B-factor (Å$^2$) | −119† | −107 | −170 | −85 |
| Original movie processing | | | | |
| Running average frames | 7 | 7 | 7 | 5 |
| CPU time (hr) | 3720 | 690 | 16,060 | 8030 |
| Resolution (Å) | 5.4 | 4.4 | 5.7 | 3.23 |
| B-factor (Å$^2$) | −199 | −166 | −228 | −76 |
| New movie processing | | | | |
| Running average frames | 7 | 7 | 7 | 5 |
| $\sigma_{NB}$ | 300 | 300 | 200 | 100 |
| CPU time (hr) | 940 | 470 | 5960 | 1300 |
| Resolution (Å) | 4.5 | 4.0 | 4.8 | 3.3 |
| B-factor (Å$^2$) | −85 | −95 | −143 | −54 |

*The molecular mass of γ-secretase is 170 kDa of protein, plus 60 kDa of disordered glycosylation. The density for the glycosylation was not visible in the electron microscopy map.

†As also explained in the main text, the γ-secretase images were first subjected to the movie processing algorithm of *Li et al. (2013)*. The resolution and B-factor reported here are after application of that algorithm, but before the original or the new particle-based approach.

Whereas the original movie processing algorithm does not improve resolution for particles smaller than 500 kDa, the new approach yields significant improvements for all four cases.

*Figure 1* shows a representative field of view for each of the four samples. The movement tracks after application of the original movie processing algorithm (but omitting rotational searches) become increasingly noisy for smaller particles, whereas for the mitoribosomes the assumption of linear movements appears to be reasonable. All four samples exhibit complex movement patterns that could not be described as a single correlated movement, but which are captured by the fitted linear tracks. The user may control two parameters to adequately model these patterns. Firstly, the width of the running averages of the movie frames to be used in the alignment of each particle will represent a compromise between how precisely movements for individual particles may be modelled, and how noisy the estimated tracks will be. Secondly, complex movement patterns on the field of view may need relatively small values of $\sigma_{NB}$, whereas too small values will lead to noise in the fitted tracks, as fewer particles are taken into account. In general, one may use running averages of fewer movie frames (i.e., less accumulated electron dose) and smaller values for $\sigma_{NB}$ for larger particles. The number of frames in the running averages can only be an odd number. For large particles such as ribosomes, a running average that accumulates approximately 5 electrons/Å$^2$ (five frames in the example below) is often a good choice. For smaller particles, running averages that correspond to

larger doses may be necessary, for example 17 electrons/Å² (seven frames) for the γ-secretase example below and 7 electrons/Å² (seven frames) for β-galactosidase and complex-I. The precise values of $\sigma_{NB}$ are less critical, in particular for relatively large particles. In practice, constructing plots like those in *Figure 1* is useful for determining suitable values for both parameters.

*Figure 2* shows the relative B-factors ($B_f$) and intercepts ($C_f$) as estimated for all four data sets, as well as the corresponding frequency-dependent weights for each movie frame. Overall, $B_f$ and $C_f$

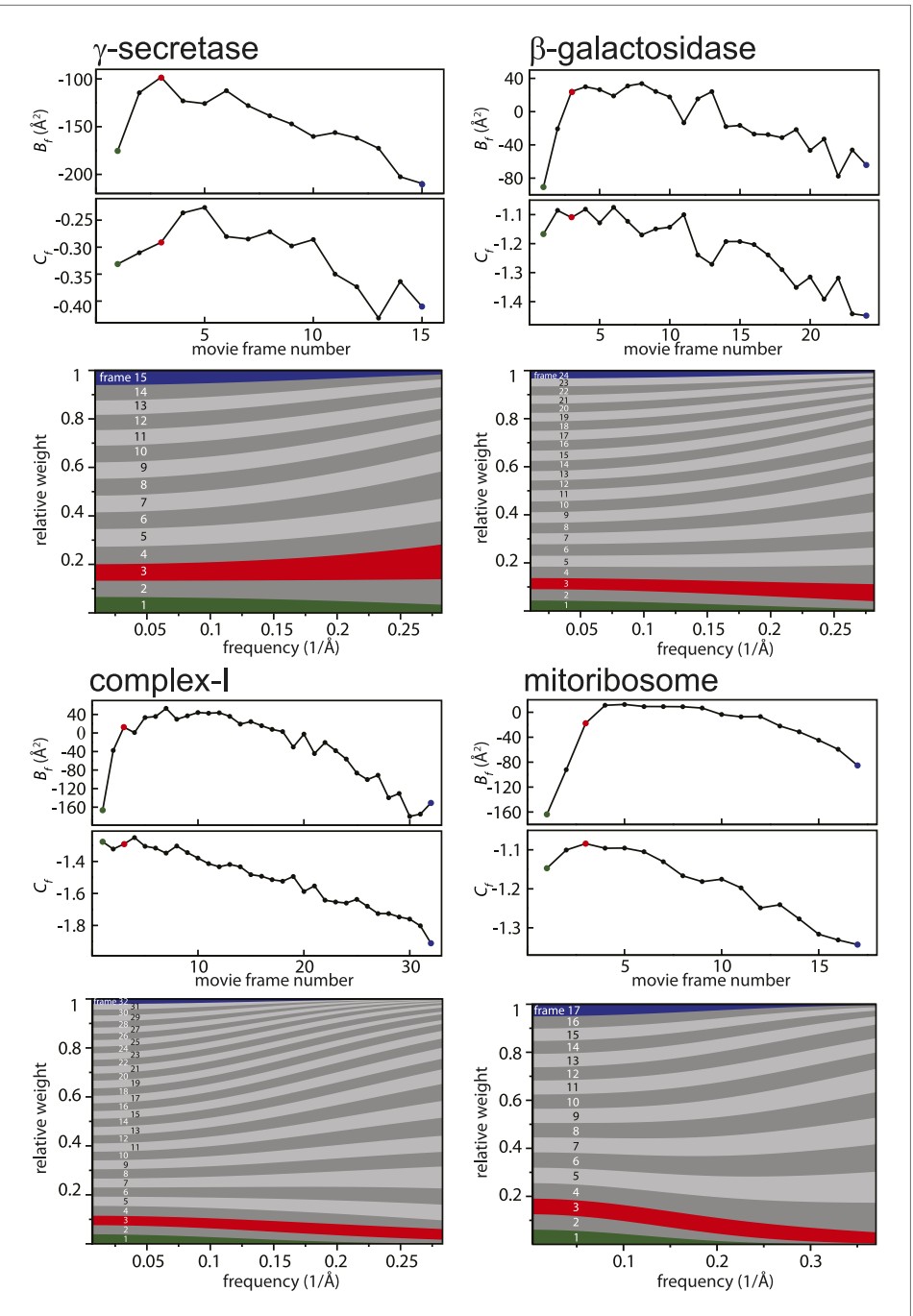

**Figure 2**. Radiation-damage weighting. For each of the four test cases, estimated values for $B_f$ and $C_f$ (top) and the resulting frequency-dependent relative weights (bottom) are shown for all movie frames. The first, third, and last movie frames of each data set are highlighted in green, red, and blue, respectively. For these movie frames, the relative Guinier plots as described in the main text and the linear fits through them are shown in *Figure 2. Continued on next page*

*Figure 2. Continued*

*Figure 2—figure supplement 1*. For example, in the γ-secretase case, the third movie frame has the least negative relative B-factor ($B_f$), and therefore this frame contributes the most of all movie frames to the weighted average at the high frequencies (and hence the red band gets broader towards the right-hand side of the relative-weight figure). In contrast, the first and last movie frames have much larger negative B-factors because they suffer from large initial beam-induced motion and radiation damage, respectively. Therefore, these movie frames contribute relatively little to the weighted average at the higher frequencies (and hence the green and blue bands decrease in width towards the right-hand side of the relative-weight figure). Because beam-induced motion and radiation damage affect the low frequencies to a much smaller extent, for the low frequencies all movie frames contribute more or less equally to the weighted average. Therefore, each band is more or less the same width on the left-hand side of the relative-weight figure, although the exact relative weights are dominated by $C_f$ on this side of the plot.

The following figure supplement is available for figure 2:

**Figure supplement 1**. Relative Guinier plots (solid lines) and the linear fits through those (dashed lines) for the first, third, and last movie frames of each data set in green, red, and blue, respectively.

values decrease with higher electron dose, which may be attributed to the relatively faster disappearance of signal at high resolution through radiation damage. Interestingly, however, consistently low values for $B_f$ are also observed for the first few movie frames. This is in agreement with observations made by others (*Brilot et al., 2012*; *Campbell et al., 2012*; *Li et al., 2013*; *Russo and Passmore, 2014*; *Vinothkumar et al., 2014a*) that during the initial stages of sample irradiation relatively large beam-induced movements occur, while sample movement slows down at higher electron doses. Given that electron diffraction spots at 3 Å resolution from bacteriorhodopsin crystals lose 99% of their intensity after a dose of only 3 electrons/Å² (*Stark et al., 1996*), the initial sample movement is extremely detrimental to high-resolution structure determination. Therefore, stopping the initial movement altogether would be a very powerful way to further increase resolution, but this has not yet been achieved with existing sample preparation or imaging techniques. Instead, the movie processing algorithm proposed here down-weights the high-frequency contributions of the first few movie frames, as well as those of the later, high-dose frames. This represents an attractive alternative to merely omitting such frames (e.g., see *Liao et al., 2013* and *Allegretti et al., 2014*), since these frames may still contribute useful information to the reconstruction process, particularly at low resolution.

*Figure 3* shows the map improvement that results from the new movie processing approach. For all four data sets the reconstructed density is significantly improved, thus allowing identification of finer biologically relevant details like separated β-strands or RNA bases in the maps. The new approach outperforms the original one for the three smaller complexes. Only for the mitoribosome LSU, does the new approach yield a map with 0.05 Å lower resolution than the original approach, although this difference is hard to appreciate in the maps (see *Figure 3—figure supplement 1*). Probably, for particles of several megadaltons in size, the determination of rotations and translations is (just about) accurate enough to follow particle rotations during the movies (also see *Brilot et al., 2012* and *Campbell et al., 2012*), and the omission of rotational searches in the new approach may actually affect the results to a small extent. (For small particles the accuracy with which such particles may be aligned is the limiting factor, and omitting rotational searches in the movie refinement will probably not have a noticeable effect.) Nevertheless, when weighed against the significant speed-up obtained by omitting the rotational searches from the original movie processing approach (a factor of 6 for the mitoribosome data set), even for large particles the new approach may still be preferred over the original one. However, the major advance of the new movie processing approach lies in its applicability to smaller particles. Whereas the original approach was limited to large particles, the new one yields map improvements for a wide range of particles, now bringing resolutions at which β-strands become separated within reach for particles smaller than 200 kDa. The new approach has been implemented in the 1.3 release of the open-source RELION program (*Scheres, 2012*), where it is called 'particle polishing'. Apart from the data sets presented here, it has been used already in the structure determination to 3.2 Å resolution of the cytoplasmic ribosome of the *Plasmodium falciparum* parasite in complex with the antibiotic emetine as well (*Wong et al., 2014*). Hopefully, in the future it will contribute positively to the structure determination of a wide range of other cryo-EM samples.

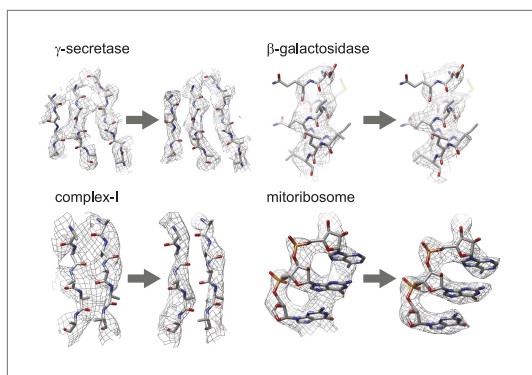

**Figure 3**. Map improvement. Representative parts of the density maps for all four test cases before (left of the arrow) and after the new movie processing approach (right of the arrow).
The following figure supplement is available for figure 3:

**Figure supplement 1**. The same part of the mitoribosome large sub-unit map as shown in **Figure 3**, but after application of the original movie processing approach, as described in **Bai et al. (2013)**.

## Materials and methods

All data sets were recorded using 300 keV electrons. The γ-secretase, β-galactosidase, and mitoribosome data sets were recorded manually; the complex-I data set was recorded automatically using the EPU software from FEI. For all data sets, fields of views that showed signs of significant drift, charging, or astigmatism were discarded. For the γ-secretase data, this assessment was made after alignment using the algorithm by *Li et al. (2013)*. Movies on the Falcon-II detectors on the Polara and Titan Krios microscopes were intercepted using a system that was developed in-house (*Bai et al., 2013*). CTF parameters were estimated using CTFFIND3 (*Mindell and Grigorieff, 2003*), and the particles were picked in a semi-automated manner, using EMAN2 (*Tang et al., 2007*) for the mitoribosome, and RELION for the three other data sets. Selection of particles for the final 3D reconstruction was performed using reference-free 2D class averaging and 3D classification in RELION (*Scheres, 2012*), and the final maps before and after movie processing were calculated using RELION's 3D auto-refine, followed by automated B-factor sharpening (*Rosenthal and Henderson, 2003*) and correction for the MTF of the detector. All resolutions were based on the gold-standard FSC = 0.143 criterion (*Scheres and Chen, 2012*), and FSC curves were corrected for the effects of soft masking by high-resolution noise substitution (*Chen et al., 2013*). Density figures were made using UCSF Chimera (*Pettersen et al., 2004*).

## Acknowledgements

I am grateful to Xiao-chen Bai for providing the mitochondrial ribosome data and the γ-secretase data; to Shaoxia Chen, Greg McMullan, and Richard Henderson for providing the β-galactosidase data; and to Vinoth Kumar for providing the complex-I data. I also thank Shaoxia Chen, Christos Savva, Jake Grimmett, and Toby Darling for technical support, and Vinoth Kumar and Richard Henderson for critical comments on this manuscript. This work was funded by the UK Medical Research Council (grant MC_UP_A025_1012).

## Additional information

### Competing interests
SHWS: Reviewing editor, *eLife*.

### Funding

| Funder | Grant reference number | Author |
| --- | --- | --- |
| Medical Research Council | MC_UP_A025_1012 | Sjors HW Scheres |

The funder had no role in study design, data collection and interpretation, or the decision to submit the work for publication.

### Author contributions
SHWS, Conception and design, Analysis and interpretation of data, Drafting or revising the article

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
