## [Decision Letter]

Thank you for sending your work entitled “Beam-induced motion correction for sub-MegaDalton cryo-EM particles” for consideration at *eLife*. Your article has been favorably evaluated by John Kuriyan (Senior editor), Werner Kühlbrandt (Reviewing editor), and 3 reviewers, one of whom, Yifan Cheng, has agreed to reveal his identity.

The Reviewing editor and the reviewers discussed their comments before we reached this decision, and the Reviewing editor has assembled the following comments to help you prepare a revised submission.

All the reviewers agreed that the manuscript represents an important advance in high-resolution cryo-EM of sub-MDa particles and should be published in *eLife* as quickly as possible. However, three reviewers raised the concern that more detail needs to be added to the description of the procedure, so that others can reproduce and apply it to their data.

Please address the following points in a revised manuscript:

1) The description of how linear tracks are fitted through particles in each frame is vague. As this is a method-oriented paper, a detailed description is necessary.

2) The first formula in the Approach section is not intuitive. It is not clear how this formula is derived. Again, a clear description is necessary.

3) Results and Discussion: “The user may control two parameters…” Presumably these are the width and sigma(NB) of the Gaussian. Can you provide some guidance how they should best be chosen?

4) Figure 1. Why are the tracks shorter and sparser for the smallest particles, and also for complex-1 relative to beta-galactosidase and ribosomes? One might expect the smaller particles to move more. Does the plot show only the second stage of correction for gamma-secretase? Some comment on this would help, even if these are just arbitrary selections from the data sets.

5) It would be good to have an idea of how many particles were grouped for tracking in each case.

6) Figure 2. The striped plots of relative weight are difficult to understand. Why are they shown as bands of variable width? Does this indicate some range of values the extent of which varies with frequency?

7) Figure 2—figure supplement 1. How can the Guinier plot increase with frequency (image 1 of beta-galactosidase)?

8) Is it possible to estimate the extent to which the resolution of smaller particles is limited by out-of-plane rotations, which are not corrected?

---

## [Author Response]

*1) The description of how linear tracks are fitted through particles in each frame is vague. As this is a method-oriented paper, a detailed description is necessary*.

The following paragraph was added:

“More specifically, for each particle *p*, the program independently minimizes the least squares target below for both the X- and Y components of the particle coordinates in the field of view:

minαp,βp(∑p'wp'∑f(Δxp'−αp−βpf)2),

where the first summation runs over all particles *p’* in the field of view; *w*_*p*_*’* is calculated as a Gaussian function of the distance between particles *p* and *p’*; the second summation runs over all movie frames *f=1,…,F*; Δ*xp’* is the difference in (either X- or Y-) coordinate between the refined position of the average particle *p’* and the refined position of it’s movie frame *f*; and α_*p*_ and β_*p*_ are the intercept and slope parameters of the fitted movement track for particle *p* in the X or Y-direction. The user controls the standard deviation σ_NB_ of the Gaussian function that defines all *w*_*p*_*’*. Smaller values for σ_NB_ result in fewer neighboring particles effectively contributing to the linear fits, so that less noise may be removed, but more complicated movement patterns may be described.”

*2) The first formula in the Approach section is not intuitive. It is not clear how this formula is derived. Again, a clear description is necessary*.

The following explanation was added above the formula:

“Instead of estimating absolute B-factors for each movie frame, the alternative approach merely estimates “relative B-factors*”* that describe how much faster (or slower) the signal in reconstructions from individual frames drops with frequency compared to the average reconstruction from all movie frames. If the signal in reconstruction from a single movie frame of all particles drops faster than the signal in a reconstruction from all movie frames then the relative B-factor will be negative; if the signal in the single-frame reconstruction drops slower the relative B-factor will be positive.”

Below the formula, the following sentence was added: “and one uses τ^2^(v)/σ^2^(v) = *SNR*(v) = *FSC*(v)/{1-*FSC*(v)}.”

Also, at the end of the Approach section, the following sentences were added:

“It is useful to consider that, since all Guinier plots are calculated relative to the same average reconstruction, the absolute values of B_f_ and C_f_ do not matter in the calculation of the weighted average particles: only the differences between these values for different movie frames determine the relative contribution of each movie frame to each frequency shell (v). Also, because of the summation in the nominator above, the sum of all w_f_(v) over all movie frames remains unity within each frequency shell. Therefore, the re-weighting of the movie-frames does not comprise an overall sharpening or dampening of the data, and the estimation of an overall absolute B-factor (as in the last row of Table 1) remains relevant.”

3) Results and Discussion: “The user may control two parameters…” Presumably these are the width and sigma(NB) of the Gaussian. Can you provide some guidance how they should best be chosen?

The following text was added:

“The number of frames in the running averages can only be an odd number. For large particles such as ribosomes, a running average that accumulates approximately 5 electrons/Å2 (5 frames in the example below) is often a good choice. For smaller particles, running averages that correspond to larger doses may be necessary, e.g., 17 electrons/Å^2^ (7 frames) for the gammasecretase example below and 7 electrons/Å^2^ (7 frames) for beta-galactosidase and complex-I. The precise values of σ_NB_ are less critical, in particular for relatively large particles. In practice, making plots like the ones in Figure 1 are useful to determine suitable values for both parameters.”

*4)*
Figure 1*. Why are the tracks shorter and sparser for the smallest particles, and also for complex-1 relative to beta-galactosidase and ribosomes? One might expect the smaller particles to move more. Does the plot show only the second stage of correction for gamma-secretase? Some comment on this would help, even if these are just arbitrary selections from the data sets.*

In the legend to Figure 1, the following text was added:

“Note that tracks are only shown for those particles that were selected for the final reconstruction after 2D and 3D classification. Also note that the relatively small movement tracks for γ-secretase only represent the beam-induced motion that was not already corrected for in the algorithm by Li et al.”

*5) It would be good to have an idea of how many particles were grouped for tracking in each case*.

An orange circle indicating the 2σ_NB_ distance was added to each of the micrographs in Figure 1.

*6)*
Figure 2*. The striped plots of relative weight are difficult to understand. Why are they shown as bands of variable width? Does this indicate some range of values the extent of which varies with frequency?*

Understanding these plots is important! The following text was added to the legend of Figure 2:

“For example in the γ-secretase case, the third movie frame has the least negative relative B-factor (B_f_), and therefore this frame contributes the most of all movie frames to the weighted average at the high-frequencies (and hence the red band gets broader towards the right-hand side of the relative-weight figure). On the contrary, the first and last movie frames have much larger negative B-factors because they suffer from large initial beam-induced motion and radiation damage respectively. Therefore, these movie frames contribute relatively little to the weighted average at the higher frequencies (and hence the green and blue bands decrease in width towards the right-hand side of the relative-weight figure). Because beam-induced motion and radiation damage affect the low frequencies to a much smaller extent, for the low frequencies all movie frames contribute more-or-less equally to the weighted average. Therefore, each band is more-or-less the same width on the left-hand side of the relative-weight figure, although the exact relative weights are dominated by C_f_ on this side of the plot.”

*7)*
Figure 2—figure supplement 1*. How can the Guinier plot increase with frequency (image 1 of beta-galactosidase)?*

The following text in the Approach section now makes this explicit:

“If the signal in reconstruction from a single movie frame of all particles drops faster than the signal in a reconstruction from all movie frames then the relative B-factor will be negative; if the signal in the single-frame reconstruction drops slower the relative B-factor will be positive.”

8) Is it possible to estimate the extent to which the resolution of smaller particles is limited by out-of-plane rotations, which are not corrected?

The following sentence was added to the Discussion:

“For small particles the accuracy with which such particles may be aligned is the limiting factor, and omitting rotational searches in the movie refinement will probably not have a noticeable effect.”